# An Interpretable Aid Decision-Making Model for Flag State Control Ship Detention Based on SMOTE and XGBoost

**Jian He** [1,2] , **Yong Hao** [1,2,*] **and Xiaoqiong Wang** [1,2]

1   School of Navigation, Wuhan University of Technology, Wuhan 430063, China;
   hj1552016868@whut.edu.cn (J.H.); wang_xq@whut.edu.cn (X.W.)
2   Hubei Key Laboratory of Inland Shipping Technology, Wuhan University of Technology,
   Wuhan 430063, China
*   Correspondence: marinehao@126.com

**Abstract:** The reasonable decision of ship detention plays a vital role in flag state control (FSC). Machine learning algorithms can be applied as aid tools for identifying ship detention. In this study, we propose a novel interpretable ship detention decision-making model based on machine learning, termed SMOTE-XGBoost-Ship detention model (SMO-XGB-SD), using the extreme gradient boosting (XGBoost) algorithm and the synthetic minority oversampling technique (SMOTE) algorithm to identify whether a ship should be detained. Our verification results show that the SMO-XGB-SD algorithm outperforms random forest (RF), support vector machine (SVM), and logistic regression (LR) algorithm. In addition, the new algorithm also provides a reasonable interpretation of model performance and highlights the most important features for identifying ship detention using the Shapley additive explanations (SHAP) algorithm. The SMO-XGB-SD model provides an effective basis for aiding decisions on ship detention by inland flag state control officers (FSCOs) and the ship safety management of ship operating companies, as well as training services for new FSCOs in maritime organizations.

**Keywords:** flag port control; ship detention decision; smart maritime; SMOTE algorithm; XGBoost

## 1. Introduction

Shipping is vital to a country's economic development, especially in respect of inland waterway transportation, which is one of the main transportation methods of goods in China. However, water traffic accidents occur frequently. A shipping accident can cause a huge loss of property, loss of life, and environmental pollution. For example, the sinking of the Eastern Star cruise ship in the middle reaches of the Yangtze River, on 1 June 2015, caused the death of 422 passengers and crew and resulted in significant property loss [1]. Therefore, reducing transport risk and avoiding shipping accidents have become increasingly important.

Flag state control (FSC) inspections provide a strong line of defense against substandard ships with serious defects, and are irreplaceable for protecting water traffic safety, especially inland watercraft navigation, and for preventing ships from causing environmental pollution [2,3]. FSC inspections are conducted by the maritime regulatory authority on ships flying the national flag in accordance with relevant laws, technical regulations, required certificates, and the ship's manning status [3]. When an FSC inspection determines that a ship has major defects affecting navigational safety, the ship's safety inspector implements mandatory measures to "detain" the ship in accordance with relevant laws, regulations, and professional knowledge. There have been a few studies on the relationship between ship defects and ship detention decisions [4–7] in previous studies. Although these methods are feasible for ship detention decision-making, they all have shortcomings. Due to the complexity and diversity of ship detention factors, and the problem of sample imbalance in FSC inspection dataset, it is necessary to combine multiple methods and use

integrated intelligent algorithms to improve decision-making accuracy. Therefore, in this study, we combine the synthetic minority oversampling technique (SMOTE) algorithm with the latest machine learning achievement, i.e., the extreme gradient boosting (XGBoost) algorithm, to make intelligent decisions on ship detention. The model eliminates the problem of sample imbalance, and ensures that the sample clearly reflects the importance of various features. Its advantages include high prediction accuracy, good fitting effect on classified data, and strong generalization ability. In addition, the interpretability analysis of this "black box" model makes the model easy to understand. Experimental results verify the reliability and practicability of the SMOTE-XGBoost-Ship detention (SMO-XGB-SD) model for making decisions on ship detention in the inland FSC inspection. Furthermore, the development of an intelligent aid for making decisions on ship detention is significant for the promotion of "smart maritime" in inland rivers.

The remainder of this paper comprises five sections. In Section 2, we review the relevant literature on FSC inspection and ship detention decision analysis. The overall framework of the SMO-XGB-SD model is presented in Section 3. In Section 4, we describe the original dataset and provide the steps of the proposed model in detail. In Section 5, we discuss the results of the analysis using the proposed model. Finally, in Section 6, we provide conclusions and further research directions.

## 2. Literature Review

This work involves two major topics, FSC ship detention analysis and XGBoost algorithm applications. The related research is reviewed below.

### 2.1. Flag Ship Control (FSC) Ship Detention Analysis

Since ship detention has an important impact on water traffic safety, many studies on ship detention have been conducted over the decades. To the best of our knowledge, based on our literature review, there are few studies on FSC ship detention, but many studies on port state control (PSC) ship detention. In addition, ship detention has been studied from two perspectives: detention factors and ship detention decision making.

In respect of detention factors, Zhang (2014) [8] selected FSC ship detention factors such as life-saving equipment, fire-fighting facilities, and navigation safety, combined with a formal safety assessment (FSA) method to evaluate the safety risk of ships by FSC inspections. Hao et al. (2016) [9] used the Apriori algorithm to conduct data mining on FSC inspection data from the Changjiang Maritime Safety Administration and showed that there were associations between ship deficiencies and FSC detention. Chen et al. (2019) [10] used a grey relational degree (GRA) analysis model with improved entropy weight to determine the key PSC ship detention factors and analyzed the degree of influence of the various factors on the ship detention decision. Tsou, M. et al. (2018) [11] used big data to analyze the relationships among ship detention deficiencies and external factors, and objectively identified regular correlations. Yang et al. (2018) [6] proposed a data-driven Bayesian network method to analyze the correlation between ship detention factors in PSC inspection and the key factors affecting ship detention, including the number of deficiencies, types of deficiencies, and age of the ship. Carious, P. et al. (2009) [12] analyzed 4080 PSC inspection reports by the Swedish Maritime Authority, from 1996 to 2001, using an econometric model, and found that age, nationality, and type of ship at the time of inspection were the main determinants in the ship detention decision. Bao et al. (2010) [13] analyzed the influence of culture on ship detention and the influence of detention rate on flag state, age of ship, inspection institution, type of ship, and recognized international organizations. Carious, P. et al. (2009) [14] analyzed the data of 515 PSC inspections from the Indian Ocean MOU region, investigated the determinants of the number of deficiencies and the possibility of detention, and finally concluded that the main factors causing ship detention were the age of the vessel and the inspection location.

In respect of ship detention decision making, Sun (2011) [5] constructed a vessel detention index system according to the principle of maximum proportion and determined

the weight of each index using the triangular fuzzy principle. Finally, a unity model of ship detention decision making was built by combining it with the fuzzy decision model. Considering the complexity and uncertainty of ship safety inspections, Zhang et al. (2020) [7] built a PSC inspection detention risk analysis model based on Bayesian network theory to determine the high-risk factors leading to ship detention and to provide an effective basis for a detention decision by a port state control officer (PSCO). Yang et al. (2018) [6] analyzed key factors affecting ship detention based on a Bayesian network, including the number of defects, types of defects, the age of the ship, etc., and developed a risk prediction tool for predicting the probability of ship detention under different circumstances, which effectively helped port authorities to rationalize their inspection regulations and check resource allocations. Kim, G. et al. (2008) [4] reported that the high detention rate of Korean ships led to their increased inspection rate in PSC countries. They established a model to identify vulnerable PSC ships through logistic regression analysis, and used the safety inspection data from 946 ships for verification. Fu, J. et al. (2020) [15] put forward a novel framework to optimize an analytic hierarchy process (AHP) model for identifying the main types of vessel defects, and introduced a simple Bayesian model for identifying the weighting of critical defects to predict the probability of ship detention. Finally, the PSC inspection dataset was used to model the performance test, and the results proved that the method could be applied to real ship safety inspection work to assist PSCOs making detention decisions, and therefore reduce the time and cost needed for PSC inspections.

*2.2. Extreme Gradient Boosting (XGBoost) Algorithm Applications*

Ship detention is essentially the classification of detention. As compared with traditional machine learning models, such as decision tree and support vector machine (SVM) [16,17], the ensemble learning model is one of the most popular concepts in machine learning, and integrates multiple weak classifiers into one strong classifier [18,19]. XGBoost, one of the most advanced integrated learning algorithms, was proposed by Chen in 2016 [20]. Since the XGBoost algorithm has the advantages of high speed, high accuracy, and good robustness, it has been applied to many fields, such as transportation safety, biomedicine, and energy manufacturing.

In the field of transportation safety, Parsa et al. (2020) [21] used the XGBoost algorithm to detect road accidents through a real-time set of data that included traffic, networks, demographics, land use, and weather characteristics. Ma et al. (2019) [22] proposed a methodology framework based on the XGBoost algorithm and a grid analysis from the perspective of city managers to study the spatial relationships between eight factors, i.e., alcohol involved, number of parties, crash type, lighting conditions, collision involvement, motorcycle collision, day of the week and time of the day, and mortality, in Los Angeles County, and accordingly provided specific recommendations on how to reduce mortality and improve road safety.

In the application of biomedicine, Bi et al. (2020) [23] developed a new interpretive machine learning approach using the XGBoost algorithm and six different types of sequential encoding schemes to distinguish m7G sites, with cross-validation showing that their approach was more accurate than other models. Mahmud et al. (2019) [24] validated the reliability and superiority of the XGBoost classifier for the determination of drug–target interactions (DTI). In the application of energy manufacturing, Wang et al. (2020) [25] proposed a brand recognition model based on SMOTE and XGBoost integrated learning in near-infrared spectroscopy (NIRS), and obtained an identification accuracy of 94.96%, which could provide a new alternative method for diesel brand recognition. However, as far as we know, based on our literature review, the XGBoost classifier has not been applied in the field of FSC ship detention decision making.

In summary, most existing studies have been aimed at the analysis of ship detention factors and decision-making models with traditional non-machine learning algorithms. With the rapid development of artificial intelligence, countries need a "smart maritime"

strategy, and therefore combining traditional ship detention theory with modern machine learning technology is a breakthrough in the field of FSC inspection.

## 3. Overall Framework

The overall framework of the SMO-XGB-SD model is illustrated in Figure 1. As shown in the figure, the model involves five procedures. First, we collected mass datasets of inland ship safety inspections. Second, for these unbalanced datasets, we used SMOTE algorithms for data preprocessing, and transformed the datasets into numeric vectors using one-hot encoding methods. Third, we built a ship detention model for inland waters using the XGBoost classification algorithm, and then continuously optimized parameters to construct the optimal SMO-XGB-SD model. Fourth, we evaluated the model's performance, and conducted comparative experiments with other algorithms. Finally, we used the Shapley additive explanations (SHAP) algorithm for the interpretability analysis of our SMO-XGB-SD model.

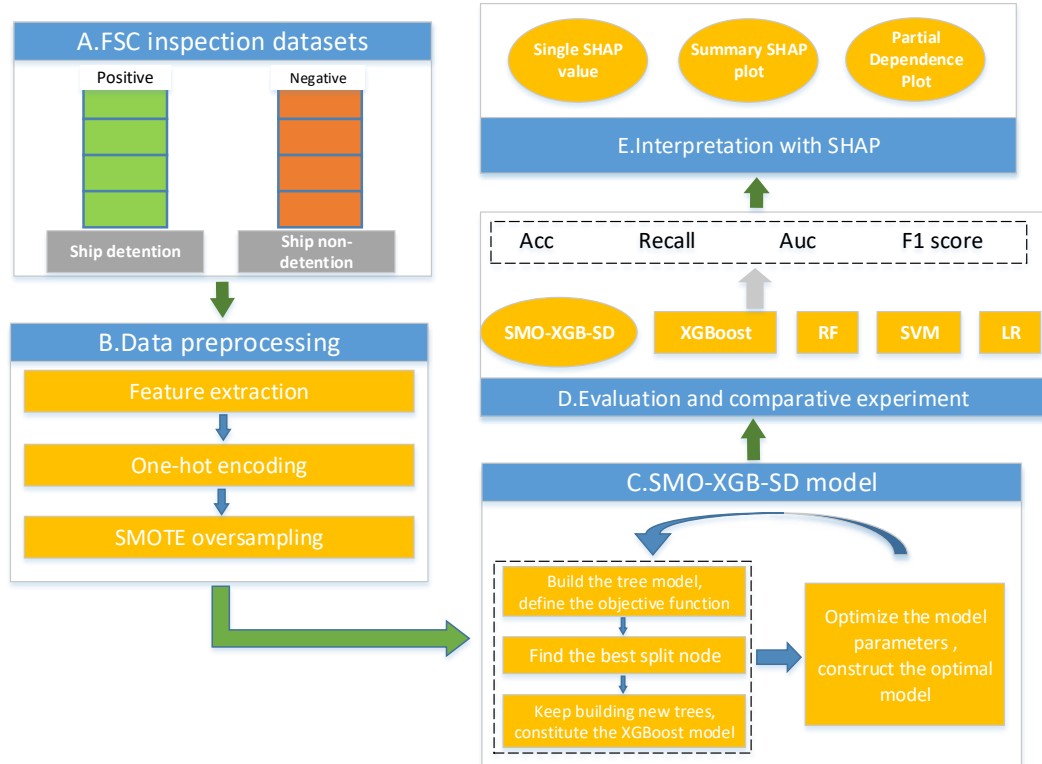

**Figure 1.** Overall framework of the ship detention decision-making model based on machine learning, i.e., the SMO-XGB-SD model.

## 4. Materials and Methods

### 4.1. Original FSC Inspection Datasets

In this study, the original datasets for training and evaluating the SMO-XGB-SD model were collected from the Changjiang Maritime Safety Administration of the People's Republic of China (CJMSA). The original datasets consisted of 75,442 FSC ship safety inspection samples and 10 features, comprising MMSI, ship's name, port of registry, date of inspection, port of inspection, inspection authority, number of deficiencies, the ship detention result, deficiency code, and defect description. The original datasets contained some sensitive ship information, which was desensitized, as shown in Table 1. The features with many missing items were first deleted from the original FSC inspection datasets. In order to further solve the problem of low precision caused by the unbalanced data, the deficiency codes were converted into major category codes (see Figure 2), and the datasets were divided into positive and negative samples, according to the ship detention results, to form the final dataset.

**Table 1.** Original flag state control (FSC) inspection datasets.

| NO. | MMSI | Ship's Name | Port of Registry | Date of Inspection | Port of Inspection | Inspection Authority | Number of Deficiencies | Ship Detention Result | Deficiency Code | Defect Description |
|---|---|---|---|---|---|---|---|---|---|---|
| 1 | 412400000 | SHIP A1 | Anqing | 23 June 2017 | Anqing | Anqing Port Marine Department | 8 | No | 9999 | Other: Port of registry and name of vessel not clear |
| 2 | 412400001 | SHIP A2 | Fuling | 16 June 2017 | Fengdu | Chongqing Fengdu Marine Department | 23 | Yes | 1499 | Other: No identification for engine-room valves |
| 3 | 412400002 | SHIP A3 | Fengjie | 15 June 2017 | Wanzhou | Chongqing Wanzhou Marine Department | 9 | Yes | 0741 | Fire hose, fittings and hydrants, hoses, squirts, one hose broken |
| 75439 | 412475438 | SHIP Z1 | Jiujiang | 1 July 2017 | Yueyang | Yueyang Linxiang Marine Department | 8 | No | 0899 | Other: No plugging equipment |
| 75440 | 412475439 | SHIP Z2 | Jiujiang | 1 July 2017 | Yueyang | Yueyang Linxiang Marine Department | 8 | No | 0830 | Pipes and wires: Engine room piping coloring does not meet the requirements |
| 75441 | 412475440 | SHIP Z3 | Jiujiang | 1 July 2017 | Yueyang | Yueyang Linxiang Marine Department | 8 | No | 9910 | National flag: defaced |

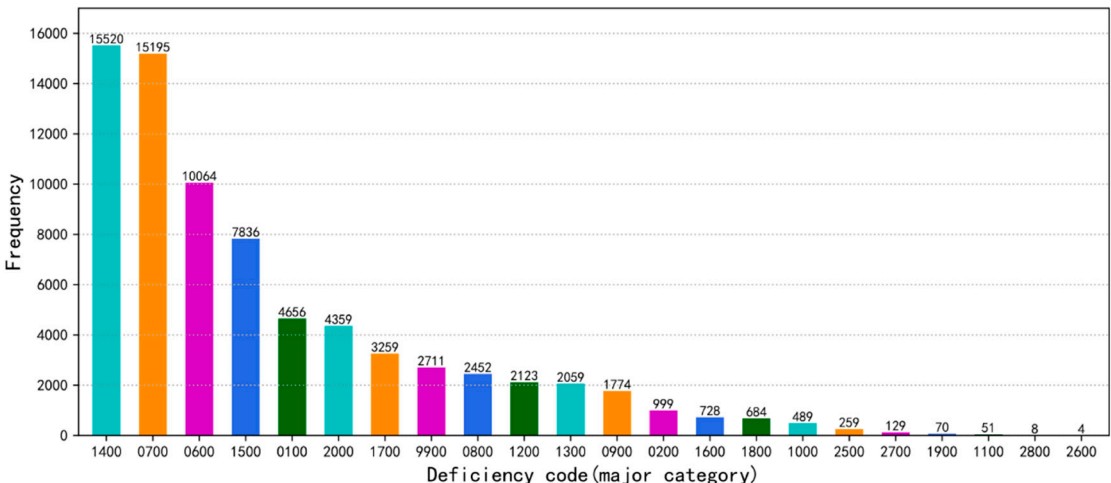

**Figure 2.** Frequency distribution of deficiency codes.

*4.2. Synthetic Minority Oversampling Technique (SMOTE)*

For this study, we selected the SMOTE method, which was proposed by Chawla [26] in 2002. It is based on the principle of oversampling the minority class and undersampling the majority class to deal with the sample imbalance problem. The class with a large number of samples is called the majority class, and the class with a small number of samples is called the minority class. When the number of samples in the minority class is too small, the accuracy of the traditional classifier is biased towards the majority class. Even if the accuracy rate is high, the classification of the minority class samples cannot be guaranteed. However, the data preprocessing technique applied to the problem of sample imbalance is different from the simple copy sample mechanism of random oversampling. The SMOTE method synthesizes new samples between two minority samples through linear interpolation, thereby effectively alleviating the overfitting problem caused by random oversampling, making the sample class distribution balanced, and improving the generalization ability of the classifier on the test set.

The basic principles of the SMOTE method [27] are as follows: firstly, select each sample $x_i$ from the minority samples as the root sample of the new synthetic sample; secondly, according to the upsampling magnification $n$, randomly select one of the $k$ neighboring samples of the same category of sample $x_i$ as the auxiliary sample for synthesizing the new sample, repeated $n$ times; then, linear interpolation is performed between the sample $x_i$ and each auxiliary sample through Equation (1), and finally $n$ synthesized samples are generated.

$$x_{new,attr} = x_{i,attr} + rand(0,1) \times (x_{ij,attr} - x_{i,attr}) \tag{1}$$

where $x_{new,attr}$ for $attr = 1, 2, \cdots, d$ is the $attr$-th attribute value of the $i$-th sample in the minority sample; $rand(0,1)$ is a random number between 0 and 1; $x_{ij}$ for $j = 1, 2, \cdots, k$ is the $j$-th nearest neighbor sample of $x_i$; $x_{new}$ represents a new sample synthesized between $x_{ij}$ and $j$.

*4.3. One-Hot Encoding*

One-hot encoding [28,29] is also called one-bit effective encoding. The method uses $N$-bit status registers to encode $N$ states. Each state has an independent register bit and, at any time, only one bit is valid. One-hot encoding is the representation of categorical variables as binary vectors with the advantage that it can transform a sample dataset into a form that is easy to use for machine learning, especially for the machine learning classification algorithm used in this study, which significantly improves the calculation speed and performance of the model.

The dates of FSC inspections were divided according to the seasons into spring, summer, autumn, and winter. According to the one-hot coding rule, each season was represented by a four-dimensional binary vector, for example, spring was coded as (1, 0, 0, 0), summer was coded as (0, 1, 0, 0), autumn was coded as (0, 0, 1, 0), and winter was coded as (0, 0, 0, 1). In addition, the FSC inspection datasets contained 22 types of major category deficiency codes, and these codes appeared multiple times in a single inspection record. For example, during the inspection on 1 July 2017, codes 0200, 0700, and 1700 appeared once, code 2000 appeared three times, and the others did not appear. These major category deficiency codes were coded as (0, 1, 1, . . . , 1, 3), whereas the omitted codes were all 0.

*4.4. XGBoost Classification Machine Learning Algorithm*

The extreme gradient boosting (XGBoost) algorithm, designed by Chen Tianqi [20], is a distributed and efficient boosting integrated classification algorithm based on decision trees. Its basic principle is to combine several low-precision weak classifiers into a high-precision classification. The XGBoost algorithm has the advantages of parallelism, high speed, and good robustness [30]. It is able to fit classification data and can automatically learn the splitting direction for missing values in the data, as well as introduce regularization and second-order Taylor expansion to improve the prediction accuracy of the algorithm. Compared with similar integrated algorithms, it has greater advantages in terms of fitting accuracy and calculation speed [31–33].

A diagram of the basic process of the XGBoost algorithm is shown in Figure 3. For a given training dataset $D = \{(x_i, y_i)\}(|D| = n, x_i \in R^m, y_i \in R)$, $n$ is the number of samples and $m$ is the number of features. According to the CART tree algorithm as the base classifier [34,35], the model function can be defined by Equation (2) as follows:

$$\hat{y}_i^{(k)} = \sum_{k}^{K} f_k(x_i) \tag{2}$$

where $K$ represents the number of decision trees, $f_k$ represents the model's $k$-th decision tree, and $x_i$ is the feature vector corresponding to sample $i$.

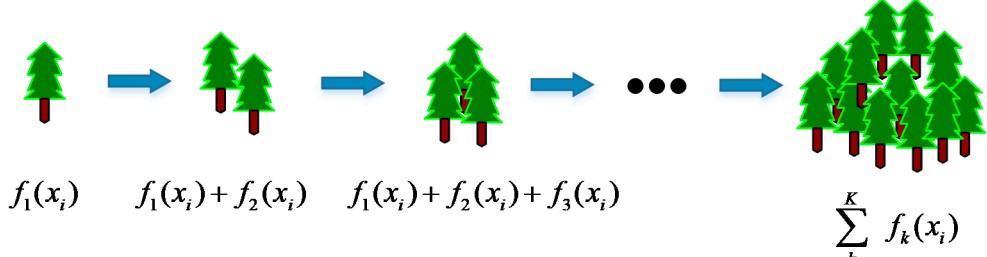

$f_1(x_i)$     $f_1(x_i) + f_2(x_i)$     $f_1(x_i) + f_2(x_i) + f_3(x_i)$        $\sum_{k}^{K} f_k(x_i)$

**Figure 3.** The process of XGBoost building tree model.

For machine learning algorithms, the core of the loss function is to measure the generalization ability of the model, that is, whether the prediction of the model on unknown data is accurate or not. XGBoost introduces model complexity to measure the computational efficiency of the algorithm; therefore, the objective function of the XGBoost algorithm is the traditional loss function plus the model complexity function, which can be written as Equation (3):

$$Obj = \sum_{i=1}^{m} l(y_i, \hat{y}_i) + \sum_{k=1}^{K} \Omega(f_k) \tag{3}$$

where *Obj* is the objective function, and $l(y_i, \hat{y}_i)$ is the training error of sample $x_i$, and $\Omega(f_k)$ is the regular term of the $k$-th classification tree.

After several rounds of iterations during the training process, the objective function of the XGBoost algorithm is expressed by Equation (4) as follows:

$$Obj^{(t)} = \sum_{i=1}^{m} l[y_i^{(t)}, \hat{y}_i^{(t-1)} + f_t(x_i)] + \sum_{k=1}^{t-1} \Omega(f_k) + \Omega(f_t) \tag{4}$$

where $f_t(x_i)$ represents the generated $t$-th classification tree and $\sum_{k=1}^{t-1} \Omega(f_k)$ represents the sum of complexity of the first $t-1$ classification trees.

Second-order Taylor approximation expansion is performed for the above formula as follows:

$$Obj^{(t)} \approx \sum_{i=1}^{m} [l(y_i^{(t)}, \hat{y}_i^{(t-1)} + f_t(x_i)g_i + \frac{1}{2}f_t^2(x_i)h_i)] + \Omega(f_t) + C \tag{5}$$

where $g_i$ and $h_i$ are the first derivative and the second derivative, respectively, with respect to $\hat{y}_i^{(t-1)}$ of the loss function $l(y_i^{(t)}, \hat{y}_i^{(t-1)})$.

The classification tree complexity is calculated by Equation (6) and, to further simplify the expression, two equations are defined as in Equation (7):

$$\Omega(f_t) = \gamma T + \frac{1}{2}\lambda \sum_{j=1}^{T} w_j^2 \tag{6}$$

$$G_j = \sum_{i \in I_j} g_i, \; H_j = \sum_{i \in I_j} h_i \tag{7}$$

Taking Equations (6) and (7) into Equation (5), we get the final objective function:

$$Obj^{(t)} \approx \sum_{j=1}^{T} \left[ w_j G_j + \frac{1}{2}w_j^2(H_j + \lambda) \right] + \gamma T \tag{8}$$

where $w_j$ is the weight of the $j$-th CART leaf node, $T$ is the number of CART leaf nodes, and $\lambda, \gamma$ are penalty coefficients.

Taking the partial derivative of objective function $Obj^{(t)}$ with respect to $w_j$ and setting the partial derivative as equal to 0 to get the optimal weight $w_i^*$:

$$w_i^* = -\frac{G_j}{H_j + \lambda} \tag{9}$$

Taking Equation (9) into Equation (8), we obtain the optimal structure of the $t$-th classification tree that minimizes the objective function:

$$Obj^{(t)} = -\frac{1}{2}\sum_{j=1}^{T} \frac{G_j}{H_j + \lambda} + \gamma T \tag{10}$$

The XGBoost algorithm uses the random subspace method when selecting the optimal split point. For each split of the node, the eigenvalues are randomly selected according to the proportion of different feature variables, and then each randomly selected eigenvalue is traversed, and the gain is selected. Choosing the split point that maximizes the gain function effectively improves the generalization ability of the model and avoids overfitting.

When selecting the split point of subtree, the gain function is defined as follows:

$$Gain = \frac{1}{2}\left[ \frac{G_L^2}{H_L + \lambda} + \frac{G_R^2}{H_R + \lambda} - \frac{(G_L + G_R)^2}{H_L + H_R + \lambda} \right] - \gamma \tag{11}$$

where $G_L$ and $H_L$ are the gradient values of the subtree on the left of the split point and $G_R$ and $H_R$ are the gradient values of the subtree on the right side of the split point.

The optimal structure and optimal split point of the new tree are determined by the above calculation, and the prediction accuracy of the model is improved by integrating the new tree.

### 4.5. Evaluation Metrics

An imbalanced data classification model cannot be evaluated using only accuracy; therefore, in this study, to evaluate the classification performance of SMO-XGB-SD we used the following evaluation metrics from multiple perspectives [36].

(1) Accuracy (Acc), precision rate (P), recall rate, and F1 score, using Equations (12)–(15), respectively:

$$Accuracy = \frac{TP + TN}{TP + TN + FP + FN} \tag{12}$$

$$Precision = \frac{TP}{TP + FP} \tag{13}$$

$$Recall = \frac{TP}{TP + FN} \tag{14}$$

$$F_1 \, score = 2 \times \frac{precision \times recall}{precision + recall} \tag{15}$$

where $TP$ originally represents a positive example and is predicted to be a positive example; $TN$ originally represents a negative example and is predicted to be a negative example; $FP$ originally represents a negative example and is predicted to be a positive example; and $FN$ originally represents a positive example and is predicted to be a negative example.

(2) Binary confusion matrix. Confusion matrix is the most basic, intuitive, and simplest method to measure the accuracy of a classification model. It separately counts the number of correct and incorrect classifications of the model, and then displays the results in a matrix, as shown in Figure 4.

**Figure 4.** The confusion matrix of binary classification.

(3) Receiver Operating Characteristic (ROC) curve and Precision Recall (PR) curve [37]. ROC curve shows the change curves of the true positive rate ($TPR$) and false positive rate ($FPR$) under different classification thresholds. PR curve shows the change curves of precision and recall under different classification thresholds. The $TPR$ and $FPR$ are defined as follows:

$$TPR = \frac{TP}{TP + FN} \tag{16}$$

$$FPR = \frac{FP}{FP + TN} \tag{17}$$

where $TPR$ is true positive rate, which is, the proportion of correctly identified positive samples in the total positive samples; $FPR$ is false positive rate, which is the actual value are negative examples and the percentage of negative examples predicted to be positive examples.

In addition, in order to make a better comparison between ROC curves, Auc, which is the area under the ROC curve, is usually used to measure the performance of a classification algorithm. The greater the value of the Auc, the better the classification performance.

### 4.6. Shapley Additive Explanations (SHAP) Method

SHAP is a model interpretation method independent of the model, which can quantify the contribution of each feature to the predictions made by the model [38]. This technique considers the impact of a single feature and the impacts of feature groups, as well as possible synergistic effects among features. The SHAP value is based on the Shapley value, which is a concept in game theory. The SHAP value of feature $i(\phi_i)$ can be computed as Equation (18):

$$\phi_i = \sum_{S \subseteq N \setminus \{i\}} \frac{|S|!(|M| - |S| - 1)!}{|M|!} [f_x(S \cup \{i\}) - f_x(S)] \tag{18}$$

where $N$ represents the set of all features in the training set and its dimension $M$; $S$ represents a permutation subset of $N$; $f_x(S)$ represents the sample average calculated using only the feature set $S$, without considering the feature $i$; $f_x(S \cup \{i\})$ represents the sample average calculated using the feature set $S$, and considering the feature $S$; $(|S|!(|M| - |S| - 1)!)/(|M|!)$ is the weight of the difference between the sample values under the feature subset $S$.

### 5. Results and Discussion

#### 5.1. Data Preprocessing and Oversampling Analysis

Data preprocessing is a very important task before establishing a model. As shown in Table 1, there are many redundant features, such as MMSI, ship name, and defect descriptions. These features have a weak correlation and a large number of missing items. Hence, the first step is to remove redundant features. A description of the five selected features is provided in Table 2.

**Table 2.** Descriptions of the selected five features.

| Feature | Value | Description | Feature | Value | Description |
|---|---|---|---|---|---|
| Location (port) of registry | AH | Anhui Province | Inspection authority (continued) | WHan | Wuhan Maritime Safety Administration |
| | CQ | Chongqing Province | | SX | Sanxia Maritime Safety Administration |
| | HEE | Henan Province | | LZ | Luzhou Maritime Safety Administration |
| | HB | Hubei Province | Number of deficiencies | 0–50 | The value range of the number of deficiencies |
| | SH | Shanghai | Deficiency code | 0100 | Ship certificate and related documents |
| | SC | Sichuan Province | | 0200 | Crew certificate and watchkeeping |
| | JX | Jiangxi Province | | 0600 | Lifesaving equipment |
| | JS | Jiangsu Province | | 0700 | Fire equipment |
| | SD | Shandong Province | | 0800 | Accident prevention |
| | ZJ | Zhejiang Province | | 0900 | Structure, stability, and related equipment |
| | YN | Yunnan Province | | 1000 | Warning signs |

**Table 2.** *Cont.*

| Feature | Value | Description | Feature | Value | Description |
|---|---|---|---|---|---|
| | LN | Liaoning Province | | 1100 | Goods |
| | GZ | Guizhou Province | | 1200 | Load line |
| Date of inspection | 1 | Spring | | 1300 | Mooring equipment |
| | 2 | Summer | | 1400 | Main power and auxiliary equipment |
| | 3 | Fall | | 1500 | Navigation safety |
| | 4 | Winter | | 1600 | Radio |
| Inspection authority | CQ | Chongqing Maritime Safety Administration | | 1700 | Dangerous goods safety and pollution prevention |
| | YB | Yibin Maritime Safety Administration | | 1800 | Oil tankers, chemical tankers, and liquefied gas tankers |
| | WH | Wuhu Maritime Safety Administration | | 1900 | Pollution prevention |
| | HS | Huangshi Maritime Safety Administration | | 2000 | Operational inspection |
| | JZ | Jingzhou Maritime Safety Administration | | 2500 | ISM/NSM |
| | YC | Yichang Maritime Safety Administration | | 2600 | Bulk carrier additional safety measures |
| | JJ | Jiujiang Maritime Safety Administration | | 2700 | Ro-ro ship additional safety measures |
| | YY | Yueyang Maritime Safety Administration | | 2800 | High-speed passenger ship additional safety measures |
| | AQ | Anqing Maritime Safety Administration | | 9900 | Others |

Subsequently, deficiency codes are converted into major category codes according to the time and location of a ship inspection, where the frequency was counted, as shown in Figure 2. Finally, the categorical variables are represented as a binary vector using one-hot encoding, and the dataset is converted to the matrix of $12401 \times 54$.

Next, the new dataset is divided into a training set and a test set at a ratio of 7:3, and SMOTE oversampling technology is used to artificially synthesize minority "detained" samples to solve the problem of imbalanced samples. The distribution of classes before and after SMOTE processing is shown in Figure 5. Figure 5a shows that the two types of sample of the original data are extremely unbalanced. In Figure 5b, we can observe that the two types of sample in the training set have reached equilibrium. Using a balanced dataset is conducive to training the classifier and can achieve higher accuracy.

*5.2. Comparison with Other Classification Algorithms*

In order to find the classification algorithm with the best performance, in this study, we selected the three most used classification algorithms, i.e., random forest (RF), support

vector machine (SVM), and logistic regression (LR) algorithm for experimental comparisons with our proposed SMO-XGB-SD model. In particular, to improve the performance of the algorithm, a grid search was used to adjust important model parameters. Table 3 shows the final parameter settings of the five machine learning algorithms.

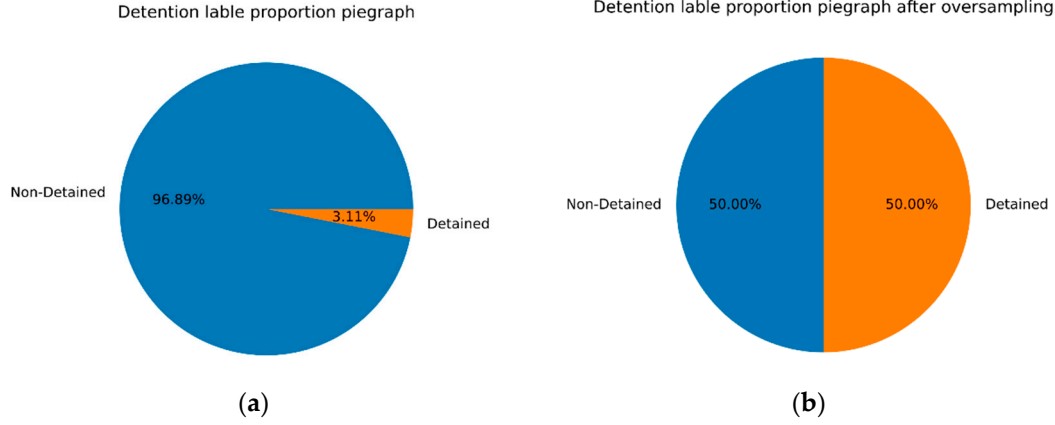

**Figure 5.** The proportions of classes (**a**) before and (**b**) after SMOTE processing.

**Table 3.** Parameter settings of SMOTE-XGBoost-Ship detention model (SMO-XGB-SD), XGBoost, random forest (RF), support vector machine (SVM), and logistic regression (LR) algorithms.

| XGBoost | SMO-XGB-SD | RF | SVM | LR |
|---------|------------|-----|-----|-----|
| booster = 'gbtree' n_estimators = 110, max_depth = 3, learning_rate = 0.3 | booster = 'gbtree' n_estimators = 110, max_depth = 3, learning_rate = 0.3 | n_estimators = 10, max_depth = 4 | C = 2, kernel = 'rbf', probability = True | C = 10, penalty = 'l2', solver = 'liblinear' |

Next, we used the final adjusted parameters to train the SMO-XGB-SD algorithm, and then used the test set for verification. The verification results are shown in Figure 6 and Table 4. According to Acc, P, Recall, and the F1 score, our proposed SMO-XGB-SD algorithm shows the best performance.

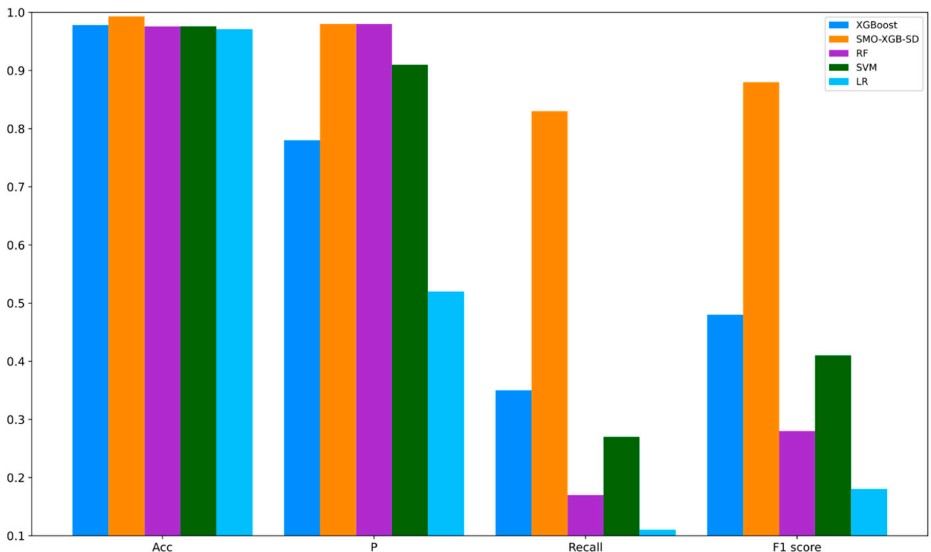

**Figure 6.** Comparison results of five classification algorithms.

**Table 4.** Comparison results of SMO-XGB-SD, XGBoost, RF, SVM, and LR.

| Models | XGBoost | SMO-XGB-SD | RF | SVM | LR |
|---|---|---|---|---|---|
| Acc | 0.978 | 0.993 | 0.976 | 0.976 | 0.971 |
| P | 0.780 | 0.980 | 0.910 | 0.910 | 0.520 |
| Recall | 0.350 | 0.830 | 0.170 | 0.270 | 0.110 |
| F1 score | 0.480 | 0.880 | 0.280 | 0.410 | 0.180 |

Furthermore, in order to visually demonstrate the prediction of the classifier, we show the binary classification confusion matrix of the five models in Figure 7. Figure 7a–e show that the color of the main diagonal gradually becomes lighter, which confirms that the number of correctly predicted "detained" ships has decreased to a certain extent. Figure 7a shows that only 22 ships were wrongly predicted as "non-detained"; the number of prediction errors for detention is significantly lower than the prediction of other models, which is a promising result for aiding FSC detention decision making. An inaccurate prediction would cause substandard ships to be missed, which could cause water traffic risks.

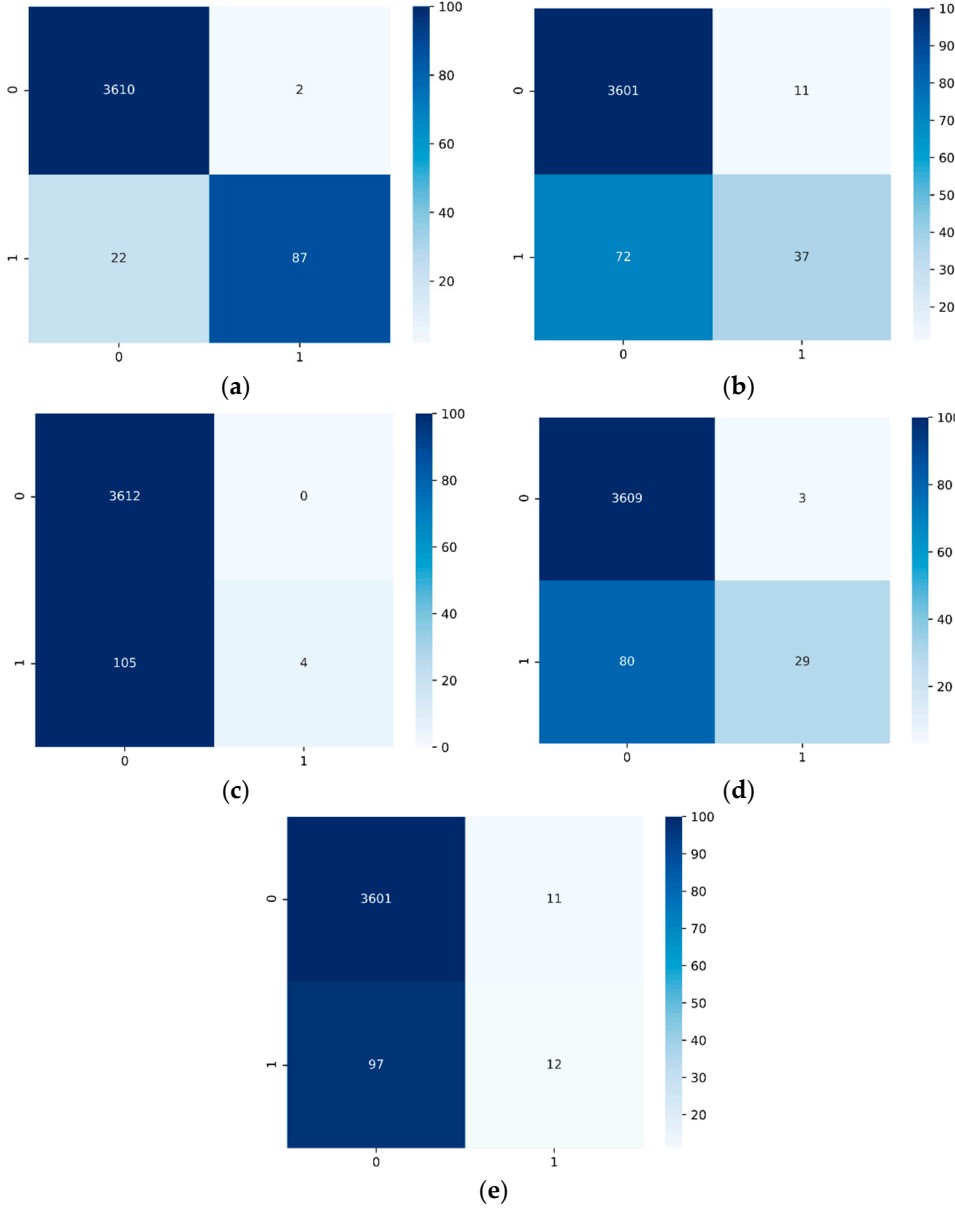

**Figure 7.** The confusion matrix of binary classification. (**a**) SMO-XGB-SD; (**b**) XGBoost; (**c**) RF; (**d**) SVM; (**e**) LR.

In order to further explore the performance of the model, an ROC curve was drawn to show the model classification effect. The closer the ROC curve overall trend is to the upper left corner, the better the model performance and the higher the probability of correctly predicting the detention class. As shown in Figure 8, the overall accuracy (Auc) of our proposed SMO-XGB-SD model is 98.7%, which is significantly higher than the other classification models. The XGBoost algorithm without SMOTE oversampling technology is 5.5% higher than LR, which reflects the superiority of the XGBoost algorithm. In addition, under the extreme imbalance of the sample dataset, the PR curve may be more practical than the ROC curve. The PR curve is different from the ROC curve, i.e., the closer to the upper right corner it is, the better the performance of the model. The model performance displayed by the PR curve is consistent with the ROC curve, as shown in Figure 9.

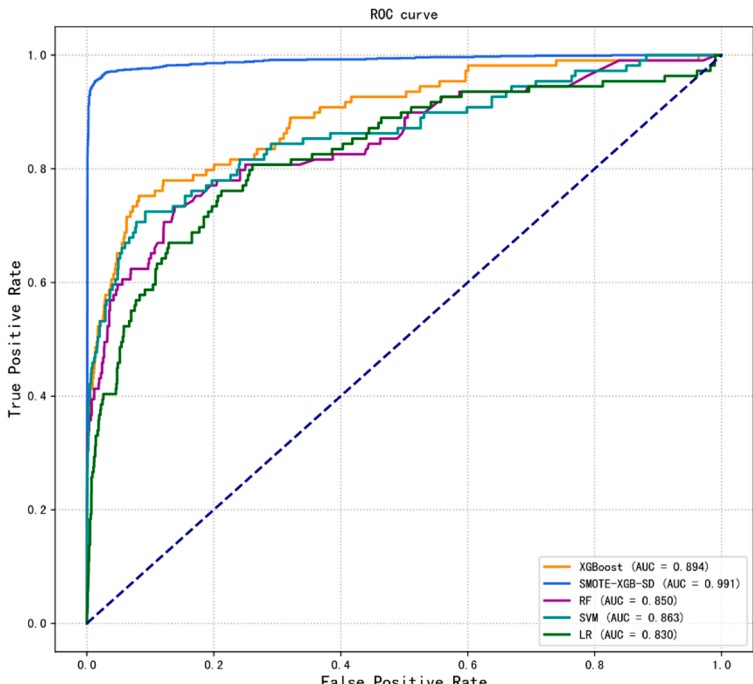

**Figure 8.** Receiver operating characteristic (ROC) curves of the five classification models.

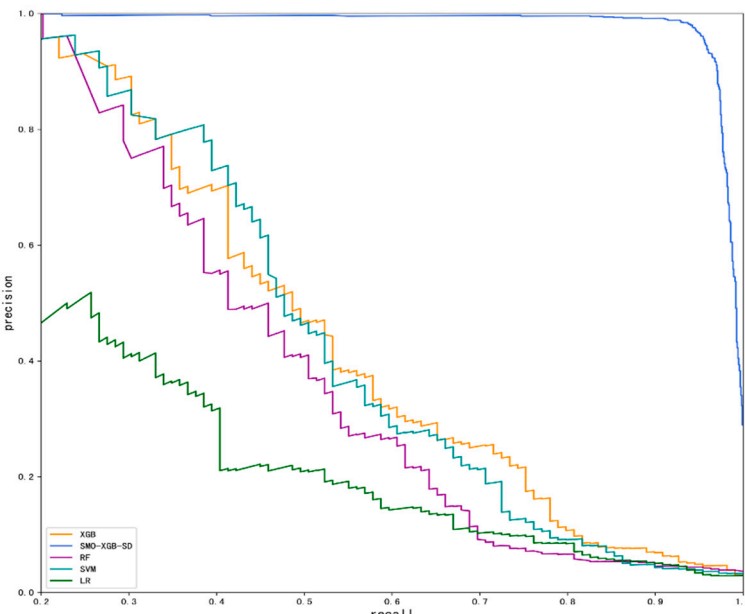

**Figure 9.** Precision recall (PR) curves of the five classification models.

### 5.3. Interpretation with SHAP Method

According to Equation (18), the SHAP value was calculated, and the top 20 features of all samples were plotted, as shown in Figure 10. In Figure 10a, the abscissa is the SHAP value, and the ordinate represents the different features. Each row represents a feature, and a point represents an FSC inspection sample. The color of the sample point indicates the size of the feature value. The redder the color, the larger the feature value; the bluer the color, the smaller the feature value. The x-coordinate value of the sample point is the influence of the feature on the model prediction of "detained". For example, the feature "0200" refers to the deficiency code of "crew certificate and watchkeeping". The redder the color of the sample point, the larger the feature value, indicating that the SHAP value is positive. The model develops towards predicting "detained" and presents a positive effect. Conversely, the bluer the color of the sample point, the smaller the feature value, indicating that the SHAP value is negative, and showing a negative effect. In an actual ship safety inspection, an inspected ship without a large number of crew certificates can easily lead to the ship being "detained".

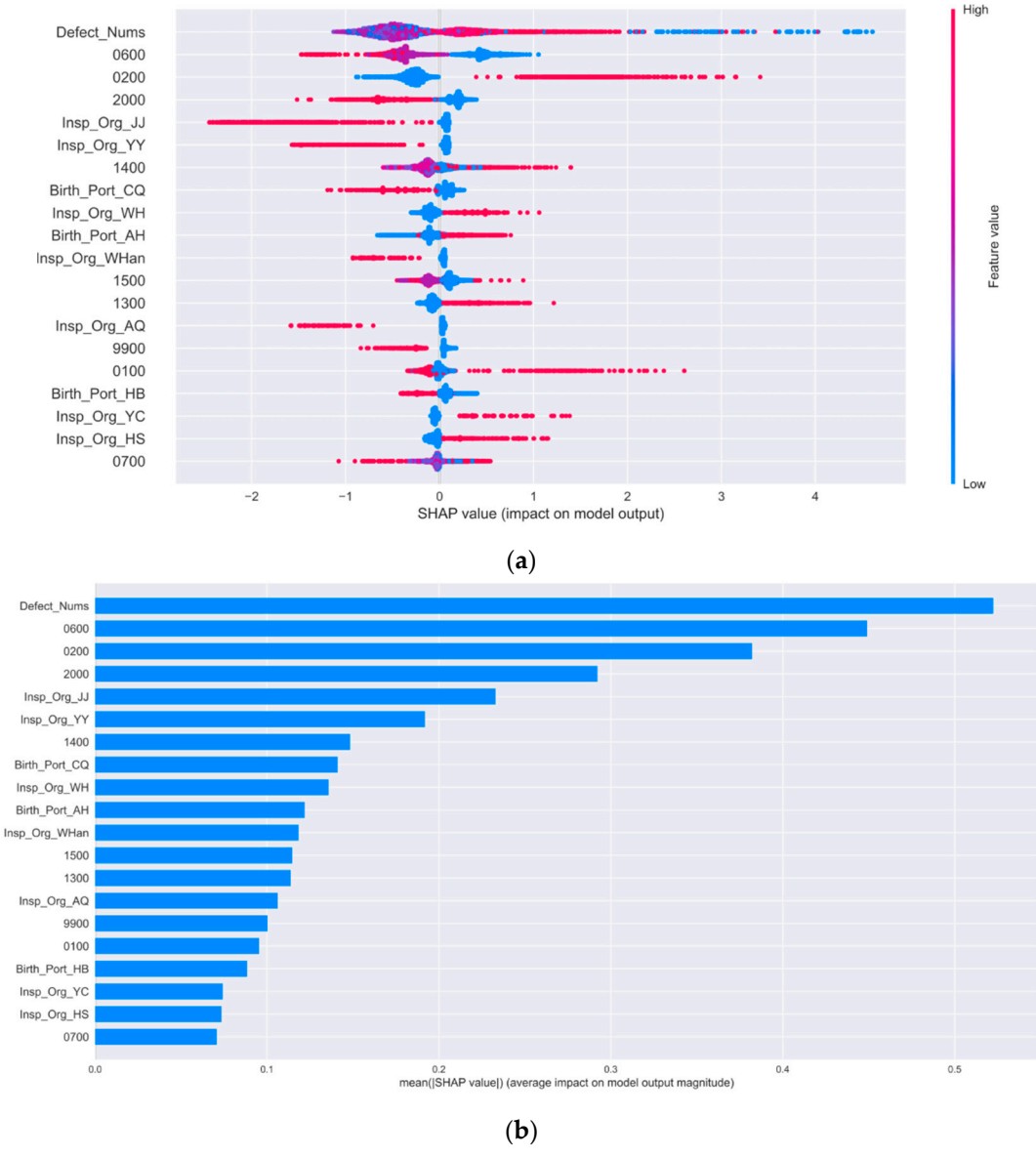

(**a**)

(**b**)

**Figure 10.** Top 20 features sorted by the Shapley additive explanations (SHAP) method. (**a**) Summary plot, SHAP values of each feature of each sample; (**b**) feature rankings based on SHAP values.

The feature ranking based on the SHAP value, shown in Figure 10b, indicates the important features that affect the model. It is consistent with the overall situation presented in Figure 10a.

## 6. Conclusions

In this study, we have proposed a novel FSC ship detention decision-making model, SMO-XGB-SD, which is used to aid flag state control officers (FSCOs) in accurately determining whether an inspected ship is "detained". Although the FSC original dataset has significant unbalanced problems, it can still accurately predict ship detention decisions. This study verified the feasibility of combining machine learning algorithms and SMOTE oversampling technology in the field of ship safety inspection. The results of a comparison of SMO-XGB-SD with other classification algorithms verifies that SMO-XGB-SD performs better in major metrics, including Acc, P, Recall, F1 score, and Auc. According to the model interpretation method SHAP, the feature contribution of SMO-XGB-SD was visually displayed and explained. In summary, the SMO-XGB-SD model proposed in this study is novel, simple, efficient, and conducive to making accurate decisions on ship detention, and therefore reduces the water traffic risk caused by substandard ships and guarantees the safety of water traffic. Moreover, it can be used to provide auxiliary training services for new employees of maritime organizations.

The scope of the FSC inspection datasets, in this study, is limited to the inland waters of the Yangtze River in China, which may not be applicable to ship detention decision making in other countries. Therefore, the adoption of big data mining and higher precision algorithms could be the focus of future work. In addition, in order to better provide guidance to stakeholders and provide auxiliary training services to maritime inspectors, the development of a ship detention decision-making aid system for mobile devices should be considered.

**Author Contributions:** Conceptualization, J.H. and Y.H.; methodology, J.H. and X.W.; validation, J.H. and Y.H.; formal analysis, Y.H.; investigation, J.H.; data curation, J.H. and Y.H.; writing—original draft preparation, J.H.; writing—review and editing, Y.H. and X.W.; visualization, J.H.; supervision, Y.H. and X.W.; project administration, Y.H. and X.W.; funding acquisition, Y.H. All authors have read and agreed to the published version of the manuscript.

**Funding:** This research was funded by the Science and Technology Support Program of the Ministry of Science and Technology grant number 2015BAG20B05 and the Changjiang Maritime Safety Administration grant number 2017h3h0374, And the APC was funded by Changjiang Maritime Safety Administration grant number 2017h3h0374.

**Institutional Review Board Statement:** Not applicable.

**Informed Consent Statement:** Not applicable.

**Data Availability Statement:** Not applicable.

**Conflicts of Interest:** The authors declare no conflict of interest.

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
