# Peer review of "An Interpretable Aid Decision-Making Model for Flag State Control Ship Detention Based on SMOTE and XGBoost"

_jmse, doi:10.3390/jmse9020156_

Round 1

Reviewer 1 Report

The paper is a study on the decision-making model of FSC and is considered a valuable study in related fields. The reviewer' opinions are as follows.

1. Abstract should be concisely and clearly described, including the background, purpose, method, result, and conclusion of the study.

2. In the description, ambiguous expressions should be avoided and quantitative figures or objective grounds should be presented. For example, this opinion can be reflected in the part describing the effects of FSC(prevention of marine accidents) and etc.

3. In this study, the composition of the dataset should be clearly explained in predicting ship defects and detentions. It should be described in such a way that general readers who understand machine learning and maritime fields can understand. In other words, it should be possible to solve the question of whether the ship's detention can be predicted by the composition of the provided dataset only.

4. As the authors have described, the methods that solved the problems perceived in previous similar studies should be described in detail(academic excellence).

5. The detention codes are the results of the FSC inspections. It would be important to be able to predict the occurrence of these codes with known or provided data. The meaning of predicting defect or pass and the definition of the prediction should be described.

In addition, it is recommended to edit/review English expressions by a native speaker or expert, and to describe the interpretation of the research results in an easy-to-understand manner.

Thank you very much.

Reviewer 2 Report

Broad comments. Using FSC data for the prediction of ship detention is a is an idea that could be used in addition to much more investigated PSC data. XGBost algorithm has been proved to be efficient computationally while provides accurate results. Furthermore, SOTE algorithm has been proved to be a very efficient optimization method for specific kid of datasets, like the ones used by authors. The authors have made a concise overview of the topic and a clear reference to existing literature. They have indicated the main task of the paper among its motivation. Finally, they have pointed out the key message and the potential benefits of their work. As a general drawback I could say that there is pure justification on the selectins made by authors regarding the classification algorithms. For example, the parameters referred in Table 3 are purely justified within the text.

Specific comments. In general, the text is very well structured and has clearly defined topics. The abstract is a very good guide for what follows. More or less all fundamental theory details that are needed are discussed and concluding remarks are sufficient. Some comments for improvement:

  1. While it is clear to most of the readers, it could be beneficial to use the acronym PSC the first time with the complete words.
  2. While existence of references to FSC ship detention are mentioned in the first paragraph of 2.1nd section, the relevant literature review considers only PSC related works.
  3. In the second line of the first paragraph of section 2.2 authors should clarify what they mean with the word decision.
  4. Similarly, in section 2.2 authors could clarify which are the eight parameters used by Mu et al (2019).
  5. Authors could refine figure 6 in order to provide quantitative results in addition to qualitative comparison of the classification algorithms.
  6. In addition, an additional figure, or a table with similar comparisons but for different parameter selection for each classification algorithm could be provided.
  7. From Figure 7 the superiority of the proposed algorithm is evident. Did authors try to perform optimization in LR model in order to optimize results. Is the same approach regarding encoding used in all models?
  8. Authors could consider renaming the Discussion section to Results and include a Discussion section with their thoughts about the way this algorithm could be used in operational environment, as mentioned in the last sentence of the abstract.
  9. Authors should follow the journal paper regarding the references within the text.

Author Response

This manuscript is a resubmission of an earlier submission. The following is a list of the peer review reports and author responses from that submission.